# Divergent Effects of Glycemic Control and Bariatric Surgery on Circulating Concentrations of TMAO in Newly Diagnosed T2D Patients and Morbidly Obese

**DOI:** 10.3390/diagnostics12112783

**Published:** 2022-11-14

**Authors:** Marina Canyelles, Antonio Pérez, Alexandra Junza, Inka Miñambres, Oscar Yanes, Helena Sardà, Noemí Rotllan, Josep Julve, José Luis Sánchez-Quesada, Mireia Tondo, Joan Carles Escolà-Gil, Francisco Blanco-Vaca

**Affiliations:** 1Institut de Recerca de l’Hospital Santa Creu i Sant Pau, Institut d’Investigacions Biomèdiques IIB Sant Pau, 08041 Barcelona, Spain; 2CIBER de Diabetes y Enfermedades Metabólicas Asociadas (CIBERDEM), 28029 Madrid, Spain; 3Department of Endocrinology and Nutrition, Hospital de la Santa Creu i Sant Pau, IIB Sant Pau, 08041 Barcelona, Spain; 4Metabolomics Platform, Department of Electronic Engineering, Universitat Rovira i Virgili, 43204 Reus, Spain; 5Department of Clinical Biochemistry, Hospital de la Santa Creu i Sant Pau, IIB Sant Pau, 08041 Barcelona, Spain; 6Department de Bioquímica i Biologia Molecular, Universitat Autònoma de Barcelona, 08041 Barcelona, Spain

**Keywords:** γ-butyrobetaine, glycemic control, liquid chromatography–mass spectrometry, obesity, trimethylamine N-oxide, type 2 diabetes

## Abstract

High circulating concentrations of the gut microbiota-derived metabolite trimethylamine N-oxide (TMAO) are significantly associated with the risk of obesity and type 2 diabetes (T2D). We aimed at evaluating the impact of glycemic control and bariatric surgery on circulating concentrations of TMAO and its microbiota-dependent intermediate, γ-butyrobetaine (γBB), in newly diagnosed T2D patients and morbidly obese subjects following a within-subject design. Based on HbA1c concentrations, T2D patients achieved glycemic control. However, the plasma TMAO and γBB concentrations were significantly increased, without changes in estimated glomerular filtration rate. Bariatric surgery was very effective in reducing weight in obese subjects. Nevertheless, the surgery reduced plasma γBB concentrations without affecting TMAO concentrations and the estimated glomerular filtration rate. Considering these results, an additional experiment was carried out in male C57BL/6J mice fed a Western-type diet for twelve weeks. Neither diet-induced obesity nor insulin resistance were associated with circulating TMAO and γBB concentrations in these genetically defined mice strains. Our findings do not support that glycemic control or bariatric surgery improve the circulating concentrations of TMAO in newly diagnosed T2D and morbidly obese patients.

## 1. Introduction

The pandemic of physical inactivity and high-caloric diets has resulted in a continued rise in obesity and type 2 diabetes (T2D) cases worldwide [1]. Cardiovascular disease is the major cause of morbidity and mortality among T2D patients [2]. Strong evidence indicates that T2D can be managed with lifestyle interventions, including exercise and diet, as well as with pharmacological therapy [1]. Furthermore, intensive glycemic control reduces the risk of future cardiovascular events and overall mortality [3].

Trimethylamine-N-oxide (TMAO) is a gut-derived metabolite mainly generated from the microbial metabolism of dietary choline and L-carnitine [4]. Trimethylamine (TMA) is generated from these precursors, absorbed in the intestine, and converted into TMAO by liver flavin monooxygenase 3 [4]. TMAO has been clinically associated with major adverse cardiovascular events and mortality [5,6,7], and experimental studies have observed TMAO exerting its proatherogenic effects through multiple pathways (reviewed in [4]). The intermediate γ-butyrobetaine (γBB) may be endogenously produced as part of the L-carnitine biosynthetic pathway or synthesized during the microbial conversion of L-carnitine to TMA [8]. γBB has also been associated with cardiovascular mortality in patients with carotid atherosclerosis [9] as well as enhanced atherosclerosis in mice, although this effect was mediated by the gut microbial metabolism of γBB to TMAO [8].

Significant evidence has shown that TMAO induces insulin resistance and T2D in experimental animals [10,11,12]. Circulating TMAO concentrations are also usually higher in patients with prevalent T2D [13,14,15]. Furthermore, two reports found that higher circulating TMAO concentrations were independently associated with the risk of future cardiovascular events and mortality in T2D patients [16,17]. However, higher TMAO concentrations have not been associated with incident diabetes in several prospective studies [18,19,20], although TMAO showed a positive association with fasting insulin concentrations in one of these reports [20]. A recent meta-analysis also showed a positive association between circulating TMAO concentrations and the risk of obesity indicated by body mass index (BMI) values [21]. In line with these findings, another study demonstrated that changes in TMAO after weight-loss diets were related to an improvement of insulin sensitivity and glucose metabolism in overweight and obese adults [22]. However, several previous prospective studies involving severely obese subjects undergoing bariatric surgery showed divergent effects on circulating TMAO concentrations [23,24,25,26]. Whether targeting the glycemic status in T2D can impact TMAO production remains unknown. Here, we aimed to evaluate whether drastic but very well-established medical improvements in HbA1c in T2D patients (glycemic control), or weight in morbid obese subjects (bariatric surgery), improve the plasma concentrations of TMAO and γBB in a within-subjects design. Furthermore, we also performed an additional study to test the effects of diet-induced obesity and insulin resistance on these gut-related metabolites in a genetically defined mice strain.

## 2. Results

### 2.1. Study Cohort Characteristics

Our study included 30 T2D patients, of whom 24 were male and 6 were female. Basal insulin was discontinued in all patients between one and four weeks after starting treatment. After insulin withdrawal, the most common pharmacological treatment was the combination of metformin and sitagliptin. Table 1 shows the clinical and biochemical parameters of the newly diagnosed T2D subjects before and after the therapeutic intervention of glycemic optimization. A significant improvement in glycemic control was obtained, as assessed via the measurement of HbAlc. This improvement in glycemic control was associated with a significant reduction in the plasma total cholesterol, LDL-C, and transaminase levels. Renal function, determined as the eGFR, was not affected. We also evaluated the clinical and biochemical parameters of 19 morbidly obese subjects (8 males and 11 females) at baseline and 12 months after bariatric surgery. Within this group, 14 (74%) patients underwent sleeve gastrectomy, and 5 (26%) patients underwent gastric bypass. As expected, bariatric surgery greatly reduced BMI, but also HbAlc, glucose, and transaminase levels, whereas it increased HDL-C; renal function was not altered (Table 1).

### 2.2. Gut-Derived Metabolites in T2D and Obese Patients

We then evaluated the baseline and follow-up concentrations of TMAO and γBB via liquid chromatography–mass spectrometry. Circulating TMAO and γBB concentrations after the glycemic control were significantly higher than their respective concentrations before glycemic optimization in new T2D patients (Figure 1). Specifically, TMAO before and after glycemic control was 9.35 mmol/L (5.7–12.95) and 14 mmol/L (7.08–21.98), respectively. In the case of γBB, concentrations were 1.35 mmol/L (1–1.6) before control and 1.5 mmol/L (1.18–1.63) afterwards.

In contrast, TMAO concentrations were not significantly affected by the bariatric surgery (10.2 µmol/L, 5.7–13.4) compared with baseline values (13.3 µmol/L, 7.4–20) in obese subjects, whereas γBB concentrations were reduced after bariatric surgery (1.2 µmol/L, 1.1–1.4) compared with those at baseline (1.5 µmol/L, 1.2–1.8) (Figure 2).

Spearman correlation parameters of delta TMAO and delta γBB, with biochemical parameters that are significantly altered in T2D patients, are shown in Table 2. Neither delta TMAO nor delta γBB were significantly correlated with any of deltas studied. In obese patients, we only found a slightly significant correlation of delta TMAO with delta ALT and delta AST, curiously in an opposite direction (Table 3).

### 2.3. Mice Experiments

Given that the Western-type diet induces obesity and insulin resistance in male C57BL/6 mice, we aimed to ascertain whether long-term feeding with the Western-type diet could affect circulating TMAO and γBB concentrations in this genetically defined mice strain [27]. As expected, the C57BL/6J mice gained weight rapidly. The HOMA-IR index also rose in mice given the Western-type diet for the period studied. However, the circulating concentrations of TMAO and γBB were not affected by obesity or insulin resistance development (Figure 3). Delta TMAO was not correlated with delta weight (Rho Spearman = −0.160; *p* = 0.646) or delta HOMA-IR (Rho Spearman = 0.258; *p* = 0.539). Additionally, Delta γBB was not associated with delta weight (Rho Spearman = −0.060; *p* = 0.871) or delta HOMA-IR (Rho Spearman = −0.313; *p* = 0.431).

## 3. Discussion

There are controversial results regarding the TMAO levels in subjects with T2D. Specifically, elevated circulating TMAO concentrations have been associated with diabetes in cross-sectional analyses [13,14] and in a retrospective case-control study [15]. In contrast, plasma TMAO concentrations were not associated with incident T2D in an observational, prospective Norwegian study [18]. Furthermore, a nested, prospective study within the framework of the PREDIMED trial found that higher baseline circulating TMAO concentrations were associated with a lower risk of T2D development [19]. To the best of our knowledge, this is the first longitudinal study that evaluates the impact of glycemic control on circulating γBB and TMAO concentrations in newly diagnosed T2D subjects. Based on data from a previous publication of our group using the same methodology [28], T2D patients presented with increased metabolite levels compared to controls; therefore, a clinical intervention should be expected to have a similar impact on their levels. However, our results revealed that glycemic control optimization not only failed to reduce these gut-microbiota-derived metabolites, but increased them. This change was initially unexpected due to the known beneficial effects of glycemic control optimization on other metabolic parameters, such as those observed in our study.

Importantly, the increased circulating concentrations of γBB and TMAO in T2D patients after glycemic control optimization were not related to alterations in the estimated glomerular filtration, a major potential confounding factor of TMAO levels [29]. The effect of glycemic control on TMAO levels could be due to different factors. Despite dietary interventions being one of the main determinants of TMAO levels in humans [30], the introduction of a healthy Mediterranean diet was not found to affect the circulating TMAO concentrations [19]. Moreover, in one study, it was reported that moderate to vigorous physical activity was inversely associated with circulating TMAO concentrations in individuals at risk of T2D, defined as impaired glucose tolerance and/or impaired fasting glycemia [31]. Overall, these findings rather indicate that healthy diet and increased physical activity would not explain the raise in circulating TMAO concentrations in newly diagnosed T2D subjects after glycemic control optimization. Therefore, the contribution of specific antidiabetic drugs to gut microbiota composition could, at least partly, explain TMAO elevations in our treated T2D subjects. Indeed, metformin can modulate the gut microbiota composition in TD2 subjects [32], enhancing potential TMA-producing bacteria [33]. Additionally, the short-term administration of metformin in subjects with newly diagnosed T2D also increased the intestinal production of glycoursodeoxycholic acid, a bile acid species, by favorably influencing gut microbiota [34]. Sitagliptin treatment was found to have no effect on the intestinal microbiota composition [35], but the treatment promoted endogenous biliary acid production [36], which is a key regulator of TMAO production [37]. Therefore, one likely explanation for the increasing concentrations of plasma TMAO in T2D patients after intensive glycemic control could be a shift toward the gut microbiota, which partly enhance its ability to generate TMAO precursors. A parallel increase in circulating γBB concentrations after glycemic control would be consistent with this hypothesis. It is, however, worth noting that the information obtained in this study is of a rather short-term nature considering the T2D evolution. Therefore, more information of the long-term results of glycemic control in T2D patients will be needed. Although this study included a small number of patients, the quite homogenous intervention used allows for a major improvement of glycemic along with limited potential confounders.

In contrast with the findings of T2D patients, bariatric surgery significantly reduced circulating γBB concentrations, but not those of TMAO, in obese subjects one year after surgery. This contrasts with the results of two previous prospective studies involving morbidly obese subjects undergoing bariatric surgery that showed higher circulating TMAO concentrations [24,25]. However, TMAO concentrations were not affected one year after gastric sleeve resection in another independent study [23]. Furthermore, another report showed reduced TMAO levels four years after bariatric surgeries, mainly sleeve resections [26]. We found a similar gut metabolite profile in patients who both underwent sleeve resection and gastric bypass, which was also independent of the presence of DM2 in these patients. The complexity of gut microbiome changes following the different bariatric surgery procedures could explain these divergent results and the lack of consistent and significant differences in our study [38].

Rodent studies show that TMAO influences obesity and glucose homeostasis [39]. Furthermore, C57BL/6 mice inoculated with *E. coli* strain Nissle 1917 and fed a high-fat diet altered *E. coli* choline catabolism, thereby increasing circulating concentrations of TMAO [40]. However, the impact of diet-induced obesity and insulin resistance on circulating TMAO concentrations has not been directly addressed in rodents. Our findings indicate that a high-fat-diet-mediated induction of obesity and insulin resistance does not affect circulating γBB and TMAO concentrations in mice.

Several limitations should also be considered in the present study. Parameters such as lifestyle or diet were not strictly registered since the study was following a clinical, real-world intervention. Therefore, their impact in TMAO and γBB levels can not be directly quantified. It should be therefore noted that we aimed to study the potential influence of glycemic optimization and bariatric surgery on both gut-related biomarkers in a daily practice scenario, in which a standarized intervention was done by highly trained clinicians based on current medical guidelines. Moreover, we did not elucidate the inner mechanism that could affect TMAO and γBB levels. It should be noted that these metabolites exhibit a complex genetic and dietary regulation that involves microbiota TMA production, liver TMAO production, and its kidney clearance.

Taken together, our results suggest that clinically validated practices, such as glycemic control in T2 patients or bariatric surgery in morbid obese patients, do not induce major reductions in TMAO plasma concentration.

## 4. Materials and Methods

### 4.1. Subjects and Biochemical Parameters

The study was a prospective follow-up study conducted in the context of clinical practice and performed following the standards for medical research in humans recommended by the Declaration of Helsinki. The study was approved by the Ethical Committee of Hospital de la Santa Creu i Sant Pau (protocol code IIBS-APO-2013-105). A total of 30 subjects newly diagnosed with T2D were enrolled in this study between 2014 and 2016. All participants—or their legally authorized representatives—provided their written informed consent to participate in a comprehensive diabetes self-management education program, which included individualized instruction regarding nutrition, physical activity, and optimized metabolic control. According to our standard protocol for the management of severe hyperglycemia, the initial therapy included triple therapy with metformin, dipeptidyl peptidase inhibitors, and basal insulin in 90% of patients. Thereafter, according to routine clinical practice, basal insulin and non-insulin drugs were modified at the discretion of the responsible physician considering the individualized glycemic targets and patient characteristics. Patients were followed up for an average period of 110 ± 43 days, with none taking lipid-lowering drugs. Nineteen morbidly obese subjects were included in the study. All subjects met the criteria used in usual clinical practice for bariatric surgery, BMI > 40 Kg/m^2^ or BMI > 35 Kg/m^2^ with comorbidities, including hypertension, diabetes, and hyperlipidemia and were free of infectious diseases, and none of them were receiving anti-obesity or anti-inflammatory drugs. The exclusion criteria were a prior diagnosis of a genetic disease affecting lipid metabolism. Obese patients underwent bariatric surgery (sleeve gastrectomy or gastric bypass) and blood samples were obtained one week before the intervention (baseline) and 12 months after.

Blood samples were collected under fasting conditions in Vacutainer^®^ tubes, and serum was obtained via centrifugation at ×10,000 *g* for 10 min, with the aliquots stored at −80 °C until analysis. The % of HbA1c was determined using a HPLC Variant II analyzer (Bio-Rad; Hercules, CA, USA) with a reference range of 4.6–5.8%. To estimate the glomerular filtration rate (eGFR), plasma creatinine levels were analyzed using an Architect c16000 analyzer (Abbott Diagnostic, Abbott Park, IL, USA). This was also the case for glucose, total cholesterol, high density lipoprotein cholesterol (HDL-C), aspartate aminotransferase (AST), and alanine aminotransferase (ALT). Low-density lipoprotein cholesterol (LDL-C) concentrations were calculated using the Friedewald equation when triglyceride concentrations were <3 mmol/L. When triglyceride concentrations were ≥3 mmol/L, ultracentrifugation was performed to separate very-low density lipoproteins (VLDL), and LDL-C was calculated considering HDL-C.

### 4.2. Mice, Diet, and the Homeostatic Model Assessment of Insulin Resistance (HOMA-IR) Determinations

All animal procedures were conducted in accordance with published regulations and reviewed and approved by the Institutional Animal Care Committee of the Institut de Recerca de l’Hospital de la Santa Creu i Sant Pau (protocol code 10626). Male C57BL/6J mice were obtained from Janvier (SC-C57J-M, Le Genest-Saint-Isle, France). At 8 weeks of age, mice were fed for 12 weeks with the Western-type diet (TD88137, Harlan Teklad, Madison, WI, containing 21% of fat and 0.2% cholesterol). Mice were kept in a temperature-controlled (22 °C) room with a 12-hour (h) light/dark cycle and food and water were provided ad libitum. Body mass was recorded once a month throughout the study. Mice were fasted for 4 h before glucose and insulin determinations. Plasma glucose concentrations were measured as described above monthly throughout the study period whereas fasting plasma insulin levels were measured using a Mouse Insulin ELISA (Mercodia, Winston Salem, NC, US). The HOMA-IR index was calculated as: [fasting serum insulin (ng/mL) × fasting serum glucose (mmol/L)]/22.5.

### 4.3. Plasma TMAO and γBB Quantification

Twenty-five µL of serum and 300 µL of acetonitrile:methanol:water (5:4:1; *v*:*v*:*v*)—containing two internal standards (IS) for quantification—were mixed and vigorously vortexed for 20 s. The internal standard was d3-methylcarnitine (d3-MeCar) for γBB and TMAO quantification. The samples were re-equilibrated on ice for 30 min and centrifuged for 10 min at 25,100× *g* in 4 °C; then, the supernatant was transferred into a specific vial prior to liquid chromatography–mass spectrometry (LC-MS) analysis. Matrix-matched calibration curves were generated using a human serum pool spiked with the standards. The concentration range of the calibration curves was 0–250 µM for TMAO and 0–25 µM for γBB.

An ultra-high-performance LC system coupled with a 6490 triple-quadrupole mass spectrometer (QqQ, Agilent Technologies) using an electrospray as an ion source (LC-ESI-QqQ) working in positive mode was used to analyze the extracts. An ACQUITY UPLC BEH HILIC column (1.7 mm, 2.1 × 150 mm, Waters) and a gradient mobile phase containing water with 50 mM ammonium acetate (phase A) and acetonitrile (phase B) were used to perform the chromatographic separation. The gradient was as follows: for 0.5 min, isocratic at 75% B; from 0.5 to 2 min, reduced to 65% B; from 2 to 2.1 min, decreased to 45% B; from 2.1 to 3.9 min, maintained at 45% B; from 3.9 to 4 min, hiked up to 75% B; and, finally, until 5.5 min, the column was equilibrated at 75% B. The flow throughout this process was 0.6 mL/min. Two milliliters of plasma extract was injected in the LC system. The mass spectrometer conditions were: a drying gas temperature of 280 °C and a sheath gas temperature of 400 °C; source and sheath gas flows of 20 and 12 L/min, respectively; a nebulizer flow of 60 psi; a nozzle voltage of 500 V; a capillary volt-age of 2500 V; and iFunnel HRF and values of 110 and 80 V, respectively. The QqQ worked in MRM mode and used predetermined transitions and collision energy (CE(V)) for TMAO, γBB, and d3-MeCar(IS) as explained in the Appendix A.

### 4.4. Statistics

It was estimated that, considering α = 0.05, power = 80%, a difference of 30% in TMAO between pairs can be detected when studying a minimum of 17 patients in each group (https://statulator.com/SampleSize/ss2PM.html# accessed on 29 April 2022). Additionally, considering that mice had genetically identical diet-induced obesity and were subjected to the same environment, we used eight male mice (https://eda.nc3rs.org.uk/experimental-design-group accessed on 29 April 2022). The anthropometric and biochemical results are presented as the median (P_25_–P_75_). A Wilcoxon matched-pairs signed rank test was performed to compare the differences between each set of matched pairs in T2D patients and morbidly obese subjects. Friedman test followed by a Dunn’s multiple comparisons test was performed in time-course analyses of mice. Correlations between variables were analyzed using Spearman’s correlation analysis and parameters were determined as delta, calculated as the value after treatment or bariatric surgery minus the value before intervention in the human studies. In the mice studies, delta was the difference between week 20 and week 12. The statistical software R (http://www.r-project.org accessed on 6 June 2022) and GraphPad Prism 6.0 software (GraphPad, San Diego, CA, USA) were used to perform all statistical analyses. A *p*-value < 0.05 was considered to represent a significant difference in all the analyses.

## 5. Conclusions

Our findings do not support that glycemic control optimization and bariatric surgery cause significantly reductions in the circulating concentrations of TMAO in T2D patients and obese subjects, respectively. Rather the contrary, glycemic control optimization increased circulating γBB and TMAO concentrations in T2D patients. The differential effects of life style interventions, drugs and surgery procedure on gut microbiota composition could explain the divergent effects of glycemic control and bariatric surgery on circulating concentrations of these gut-related metabolites in T2D patients and morbidly obese subjects. More research studies are needed to evaluate the specific mechanisms by which glycemic control therapies promote these microbiota-dependent metabolites in T2D patients.

## Figures and Tables

**Figure 1 diagnostics-12-02783-f001:**
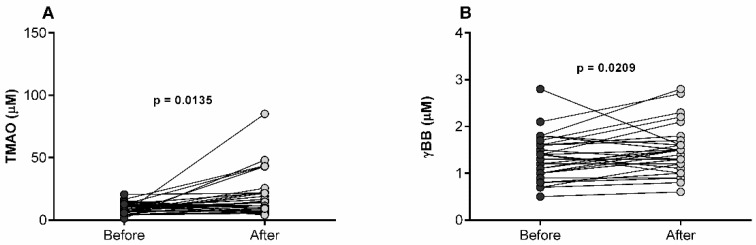
Serum TMAO (**A**) and γBB (**B**) levels in type 2 diabetic subjects at baseline and follow-up. Each pair of symbols with a line represents a participant before and after glycemic control. Wilcoxon matched-pairs signed rank test was performed to compare data.

**Figure 2 diagnostics-12-02783-f002:**
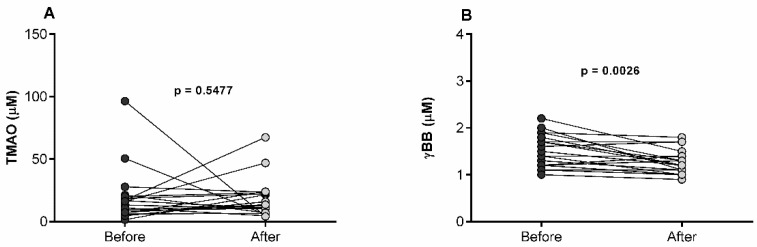
Serum TMAO (**A**) and γBB (**B**) levels in morbidly obese subjects at baseline and one year after bariatric surgery. Each pair of symbols with a line represents a participant before and after bariatric surgery. Wilcoxon matched-pairs signed rank test was performed to compare data.

**Figure 3 diagnostics-12-02783-f003:**
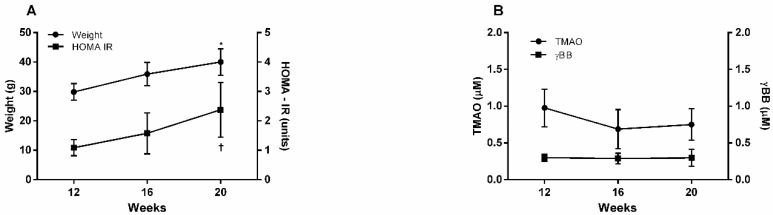
Weight and HOMA-IR (**A**) and serum TMAO and γBB (**B**) levels in C57BL6/6J mice fed with a Western-type diet. A Friedman test followed by a Dunn’s multiple comparisons test were performed in time-course analyses of mice. * *p* = 0.0002 (weight at 12 vs. 20 weeks) † *p* = 0.0005 (HOMA-IR at 12 vs. 20 weeks).

**Table 1 diagnostics-12-02783-t001:** Characteristics of T2D patients before and after glycemic optimization, and obese subjects before and after bariatric surgery.

	T2D Patients (n = 31)		*p* Value	Obese Patients (n = 19)		*p* Value
	Before	After		Before	After	
Age (y)	52 (44–57)		-	55 (47–61)		-
Sex (M/F)	25/6		-	8/11		-
BMI (Kg/m^2^)	28.9 (25.8–32.2)	28.7 (25.7–31.8)	0.0514	42.4 (39.9–49.1)	30.2 (28.5–34.8)	<0.0001
HbA1c (%)	11.2 (10.5–12.3)	5.85 (5.48–6.63)	<0.0001	5.7 (5.3–6.3)	5.2 (5–5.7)	0.0037
Glucose (mmol/L)	13.7 (12–17.9)	5.85 (5.18–6.43)	<0.0001	5.8 (5.3–7.2)	4.9 (4.7–5.5)	0.0004
Total cholesterol (mmol/L)	6.21 (5.23–7.03)	5.19 (4.61–5.67)	<0.0001	5.17 (4.77–5.52)	4.79 (4.39–5.55)	0.5678
HDL-C (mmol/L)	0.98 (0.9–1.12)	1.04 (0.92–1.23)	0.1116	1.17 (1.05–1.25)	1.43 (1.17–1.49)	<0.0001
LDL-C (mmol/L)	3.98 (3–5.04)	3.2 (2.84–3.82)	<0.0001	3.34 (2.78–3.81)	3.33 (2.57–3.69)	>0.9999
eGFR—(CKD-EPI) (mL/min/1.73 m^2^)	90 (86.4–90)	90 (90–90)	0.2061	90 (85.9–90)	90 (90–90)	0.0781
ALT (IU/L)	29 (19.5–51)	16 (13.8–26)	0.0012	26 (18–31)	13 (11–18)	0.0004
AST (IU/L)	20 (16–42.5)	18 (14–21.5)	0.0010	21 (14–27)	14 (13–21)	0.0206

BMI: body mass index; HbA1c: hemoglobin A1c; HDL-C and LDL-C: high- and low-density lipoprotein cholesterol; eGFR: estimated glomerular filtration rate by CKD-EPI equation; ALT: alanine aminotransferase; AST: aspartate aminotransferase. Results are presented as medians (P_25_–P_75_).

**Table 2 diagnostics-12-02783-t002:** Characteristics of T2D patients before and after glycemic optimization, and obese subjects before and after bariatric surgery.

	Delta HbA1c	Delta Glucose	Delta Total Cholesterol	Delta LDL-C	Delta ALT	Delta AST
**Delta TMAO**	0.13 (−0.26 to 0.49) *p* = 0.4914	0.001 (−0.37 to 0.37) *p* = 0.9953	−0.13 (−0.48 to 0.25) *p* = 0.4804	−0.12 (−0.47 to 0.26) *p* = 0.5271	0.24 (−0.14 to 0.56) *p* = 0.2041	0.20 (−0.19 to 0.53) *p* = 0.3016
**Delta γBB**	−0.01 (−0.38 to 0.37) *p* = 0.9603	−0.032 (−0.40 to 0.34) *p* = 0.8650	0.15 (−0.23 to 0.49) *p* = 0.4337	0.24 (−0.14 to 0.56) *p* = 0.194	0.31 (−0.07 to 0.61) *p* = 0.0952	0.01 (−0.36 to 0.38) *p* = 0.9696

All parameters are presented as delta, calculated as the value after glycemic control minus the value before glycemic control. Data are presented as Rho Spearman (95% confidence intervals) and have a *p* value.

**Table 3 diagnostics-12-02783-t003:** Correlation parameters of delta TMAO and delta γBB with biochemical parameters in obese patients.

	Delta BMI	Delta HbA1c	Delta Glucose	Delta HDL-C	Delta ALT	Delta AST
**Delta TMAO**	0.40 (−0.08 to 0.73) *p* = 0.088	−0.15 (−0.57 to 0.34) *p* = 0.5447	−0.36 (−0.70 to 0.13) *p* = 0.1314	0.39 (−0.09 to 0.73) *p* = 0.0973	−0.48 (−0.77 to −0.02) *p* = 0.036	0.46 (−0.003 to 0.76) *p* = 0.0456
**Delta γBB**	0.037 (−0.44 to 0.49) *p* = 0.8803	−0.24 (−0.64 to 0.25) *p* = 0.3132	−0.14 (−0.57 to 0.34) *p* = 0.551	0.15 (−0.34 to 0.58) *p* = 0.5268	−0.22 (−0.63 to 0.27) *p* = 0.3548	0.24 (−0.26 to 0.63) *p* = 0.3308

All parameters are presented as delta, calculated as the value after bariatric surgery minus the value at baseline. Data are presented as Rho Spearman (95% confidence intervals) and have a *p* value.

## Data Availability

The data that support the findings of this study will be available to other researchers upon reasonable request.

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
