# Peer review of "Divergent Effects of Glycemic Control and Bariatric Surgery on Circulating Concentrations of TMAO in Newly Diagnosed T2D Patients and Morbidly Obese"

_diagnostics, 2022, doi:10.3390/diagnostics12112783_

Round 1

Reviewer 1 Report

[1]. The abstract is a bit muddled, mixing a clinical study with an experimental one.

[2]. This reviewer's main concern is that the authors assess many parameters, without producing neither an a priori sample size calculation nor a post hoc attained power calculation. Such an addition would add to the credibility of an - otherwise - negative study.

Author Response

Reply to reviewer 1:

We thank the reviewer for her/his comments and suggestions, which have helped us improve the manuscript. Our responses are presented below, and the changes have been highlighted in the revised version of the manuscript.

1) The abstract is a bit muddled, mixing a clinical study with an experimental one.

We appreciate your comment about the abstract and we have rewritten it in order to be more understandable. We first performed the prospective follow-up study and in order to sustaine our results we also performed an additional study in a genetically-defined strain of mice that develops insulin resistance and obesity under a Western-type diet.

2) This reviewer's main concern is that the authors assess many parameters, without producing neither an a priori sample size calculation nor a post hoc attained power calculation. Such an addition would add to the credibility of an - otherwise - negative study.

It was estimated that, considering α=0.05, power=80%, a difference of 30% in TMAO between pairs can be detected when studying a minimum of 17 patients in each group (https://statulator.com/SampleSize/ss2PM.html#). Considering that mice had genetically identical diet-induced obesity and were subjected to the same environment, we used eight male mice (https://eda.nc3rs.org.uk/experimental-design-group). These points have now been added to the Statistical analyses section.

Reviewer 2 Report

This paper evaluates the impact of glycemic control and bariatric surgery on circulating concentrations of TMAO and γBB in newly diagnosed T2D patients and morbidly obese subjects. The topic is interesting and clinically significant; however, some methodological limitations are non-negligible. I recommend that this paper not be accepted without major revision.

Major comments:

1. Clinical trials require pre-registration. Please provide the registration information.

2. The clinical trials in this study have no control groups, which weakens the strength of the evidence.

3. Please provide details for the sample size calculation.

4. For glycemic control, the treatment duration of 4 weeks maybe not be long enough to achieve a solid conclusion. Is there any rational basis for setting the intervention duration as four weeks?

5. The intervention of the present study is rather crude; the parameters such as lifestyle, diet structure, and bedtime routine have not been controlled, which may influence the levels of TMAO and γBB. Therefore, this study design is not rigorous.

6. The exploration of this study lacks in-depth. It only provides simple observation data; however, the inner mechanism affecting the TMAO metabolism has not been investigated. For instance, the gut microbiota should be considered to make this study more complete.

Author Response

Reply to reviewer 2:

We acknowledge the reviewer for her/his comments and suggestions which have helped us to significantly improve the manuscript. Our responses are presented below, and the changes have been highlighted in the revised version of the manuscript.

This paper evaluates the impact of glycemic control and bariatric surgery on circulating concentrations of TMAO and γBB in newly diagnosed T2D patients and morbidly obese subjects. The topic is interesting and clinically significant; however, some methodological limitations are non-negligible. I recommend that this paper not be accepted without major revision.

Major comments:

1) Clinical trials require pre-registration. Please provide the registration information.

There is no pre-registration information because the study was a prospective follow-up study conducted in the context of clinical practice (protocol code IIBS-APO-2013-105, approved by the Ethical Committee of Hospital de la Santa Creu i Sant Pau). All participants—or their legally authorized representatives—provided their written informed consent to participate in the study.

2) The clinical trials in this study have no control groups, which weakens the strength of the evidence.

It may be worth to emphasize that the samples studied correspond to patients treated in real clinical practice. This experiment used a within-subjects design and, therefore, no control groups exist in the context of a therapeutic intervention to optimize glycemic control or bariatric surgery. Our aim was to study the impact of glycemic control and bariatric surgery on TMAO and γBB concentrations since both metabolites were linked with cardiometabolic risk. This point has now been clarified at the end of the introduction section and the discussion section.

In a previous publication of our group using the same methodology, both T2D and obese patients presented with increased metabolites levels compared to controls, therefore, a clinical intervention should be expected to have a similar impact on their levels (Canyelles, M., et al. TMAO and Gut Microbial-Derived Metabolites TML and γBB Are Not Associated with Thrombotic Risk in Patients with Venous Thromboembolism. Journal of clinical medicine, 11(5), 1425. (https://doi.org/10.3390/jcm11051425).

3) Please provide details for the sample size calculation.

It was estimated that, considering α=0.05, power=80%, a difference of 30% in TMAO between pairs can be detected when studying a minimum of 17 patients in each group (https://statulator.com/SampleSize/ss2PM.html#). Considering that mice had genetically identical diet-induced obesity and were subjected to the same environment, we used eight male mice (https://eda.nc3rs.org.uk/experimental-design-group). These points have now been added to the Statistical analyses section.

4) For glycemic control, the treatment duration of 4 weeks maybe not be long enough to achieve a solid conclusion. Is there any rational basis for setting the intervention duration as four weeks?

Patients were followed up for an average period of 110 ± 43 days, corresponding to 3 – 4 months. Since glycemic control was assessed based on HbA1c levels (its average lifetime is lower than 3 months), we believe that our follow-up was robust enough to evaluate the link between microbiota-related metabolites and glycemic control.

5) The intervention of the present study is rather crude; the parameters such as lifestyle, diet structure, and bedtime routine have not been controlled, which may influence the levels of TMAO and γBB. Therefore, this study design is not rigorous.

Thank you for bringing this important point to our attention.  We included a paragraph of limitations in the Discussion. It is known that diet and lifestyle could have an influence on gut-related metabolites such as TMAO and γBB, although not all studies confirmed this relationship. The extent to which lifestyle and diet structure influence the observed changes in both metabolites cannot be elucidated from our study. However, we rather aimed to study the use of both biomarkers in a daily clinical practice scenario, in which a strict control of diet and lifestyle are recommended. After these interventions major changes in glucose, HbA1c and lipids, but not in TMAO, were identified. We feel that we can conclude that clinically induced glycemic control in type 2 patients and bariatric surgery in obese patients (with or without diabetes) do not produce major improvements in plasma TMAO.

6) The exploration of this study lacks in-depth. It only provides simple observation data; however, the inner mechanism affecting the TMAO metabolism has not been investigated. For instance, the gut microbiota should be considered to make this study more complete.

We really appreciate the reviewer comment about this topic and included this and other factors controlling TMAO concentrations in the limitation’s paragraph of Discussion. Indeed, TMAO exhibits a complex genetic and dietary regulation that involves not only microbiota TMA production but also, liver TMAO production and its kidney clearance. This study was not oriented to provide mechanistic information but rather to answer a clinically relevant question. Does the usual clinical glycemic control in newly diagnosed T2D patients and bariatric surgery in obese patients improve TMAO and/or γBB levels? Gut-related metabolites were associated with both cardiometabolic diseases and we aimed to explore how effective treatments could impact these metabolites. Study of molecular mechanisms may be of interest in the future but it is difficult to believe  that they could change proved live-saving clinical strategies such as glycemic control in T2D and bariatric surgery in morbid obese.

Round 2

Reviewer 2 Report

The authors have revised the manuscript comprehensively. Although not every issue I pinpointed has been well addressed, however, I think it is acceptable, for some ideal designs are difficult to implement. Thank the authors for their kind revisions, and wish them better achievements in future scientific exploration.